# Subspecies in the global human gut microbiome

Paul I Costea[1] [iD], Luis Pedro Coelho[1], Shinichi Sunagawa[1,2], Robin Munch[1], Jaime Huerta-Cepas[1], Kristoffer Forslund[1], Falk Hildebrand[1], Almagul Kushugulova[3], Georg Zeller[1,*] [iD] & Peer Bork[1,4,5,6,**] [iD]

## Abstract

Population genomics of prokaryotes has been studied in depth in only a small number of primarily pathogenic bacteria, as genome sequences of isolates of diverse origin are lacking for most species. Here, we conducted a large-scale survey of population structure in prevalent human gut microbial species, sampled from their natural environment, with a culture-independent metagenomic approach. We examined the variation landscape of 71 species in 2,144 human fecal metagenomes and found that in 44 of these, accounting for 72% of the total assigned microbial abundance, single-nucleotide variation clearly indicates the existence of sub-populations (here termed subspecies). A single subspecies (per species) usually dominates within each host, as expected from ecological theory. At the global scale, geographic distributions of subspecies differ between phyla, with Firmicutes subspecies being significantly more geographically restricted. To investigate the functional significance of the delineated subspecies, we identified genes that consistently distinguish them in a manner that is independent of reference genomes. We further associated these subspecies-specific genes with properties of the microbial community and the host. For example, two of the three *Eubacterium rectale* subspecies consistently harbor an accessory pro-inflammatory flagellum operon that is associated with lower gut community diversity, higher host BMI, and higher blood fasting insulin levels. Using an additional 676 human oral samples, we further demonstrate the existence of niche specialized subspecies in the different parts of the oral cavity. Taken together, we provide evidence for subspecies in the majority of abundant gut prokaryotes, leading to a better functional and ecological understanding of the human gut microbiome in conjunction with its host.

**Keywords** genetic variation; metagenomics; microbiome; population structure; prokaryotic subspecies
**Subject Categories** Genome-Scale & Integrative Biology; Microbiology, Virology & Host Pathogen Interaction
**Mol Syst Biol. (2017) 13: 960**

## Introduction

Despite the long-standing debate on whether there is a coherent species concept in the prokaryotic world (Achtman & Wagner, 2008; Doolittle & Zhaxybayeva, 2009), modern molecular technologies offer multiple operational definitions of species, as well as of subordinate levels, such as ecotypes, that are being successfully applied to aid the discovery process in microbial ecology. Some rely on genome-wide features, such as DNA–DNA hybridization (Stackebrandt *et al*, 2002) or pairwise average nucleotide identity (Konstantinidis & Tiedje, 2005), while others are restricted to a specific marker gene (e.g., 16S rRNA gene, or a variable region therein) or multiple ones (Mende *et al*, 2013), (i.e., multilocus sequence typing). These schemes aim to classify prokaryotic strains or isolates [which may differ by just a single nucleotide (Viana *et al*, 2015)] into genetically or phenotypically coherent taxa; these either delineate a specific feature of a population (e.g., antibiotic resistance or invasion potential) or are determined in an unsupervised manner following observed population structure (Urwin & Maiden, 2003). Noting such consistent differences within operationally defined species has prompted the introduction of the subspecies concept as early as the 1950s, when *Bacillus cereus* could be stratified into virulent and non-virulent types (Davenport & Smith, 1952). Subspecies showing adaptation to different environments are also referred to as "ecotypes", a prominent example being the ocean-dwelling *Prochlorococcus* spp. (Johnson *et al*, 2006), for which specific subspecies are specialized for different environmental conditions (Biller *et al*, 2014). In regard to the human microbiome, extensive work has been undertaken to characterize population structure and describe subspecies in species that include pathogenic bacteria, such as *Escherichia coli*, different *Salmonella* species, and a few others, for which multiple typing schemes exist (Gordienko *et al*, 2013; Chakraborty *et al*, 2015; Bale *et al*, 2016; Sharma *et al*, 2016), but not for the vast majority of human gut commensals.

Current genomic approaches to determine population structure are hampered by their dependence on bacterial isolates and sequenced genomes. Due to limited throughput and biased by growth requirements, cultivation-based assessments are unviable

1 Structural and Computational Biology Unit, European Molecular Biology Laboratory, Heidelberg, Germany
2 Department of Biology, Institute of Microbiology, ETH Zurich, Zurich, Switzerland
3 Center for Life Sciences, Nazarbayev University, Astana, Kazakhstan
4 Max-Delbrück-Centre for Molecular Medicine, Berlin, Germany
5 Molecular Medicine Partnership Unit, Heidelberg, Germany
6 Department of Bioinformatics, Biocenter, University of Würzburg, Würzburg, Germany
*Corresponding author. Tel: +49 6221 387 8361; E-mail: zeller@embl.de
**Corresponding author. Tel: +49 6221 387 8361; E-mail: bork@embl.de

for the study of complex microbial communities. Recently, the advent of culture-independent metagenomic approaches and newly developed analysis tools have made it possible to study the population structure and genomic variation of microbes in their natural environment (Luo *et al*, 2015; Nayfach *et al*, 2016; Truong *et al*, 2017). Here, we devised a novel approach for broadly delineating subspecies in the majority of abundant gut microbes and determined their associated single-nucleotide variants in order to quantify and characterize them in their natural habitat. We observed consistent functional differences between subspecies with respect to their gene pools suggesting that they likely differ in phenotypic or ecological properties. We discovered that clearly delineated subspecies are the rule rather than the exception in the gut microbiome. When analyzing their global geographic distribution, we found that within a host, they generally persist stably over time and are mutually exclusive. Illustrating the utility of the subspecies concept, we associated particular subspecies with specific genes and host phenotypes, with implications for disease.

## Results

To be able to determine population structure without relying on limited or biased sets of pan-genomes (Gordienko *et al*, 2013), we explored the natural genomic variation of the human gut microbiome (Schloissnig *et al*, 2013; Luo *et al*, 2015; Nayfach *et al*, 2016; Lloyd-Price *et al*, 2017; Truong *et al*, 2017). To this end, we assessed the microbial genetic landscape of 2,144 deeply sequenced human stool metagenomes from nine countries, spanning three continents, including published (Huttenhower *et al*, 2012; Qin *et al*, 2012; Karlsson *et al*, 2013; Le Chatelier *et al*, 2013; Zeller *et al*, 2014) as well as 298 newly generated ones (Table EV1). The newly sequenced samples include a Kazakh cohort as well as three individuals from Germany that have been sampled over an extended period

of time (see Materials and Methods). By mapping the respective reads to a set of 1,753 previously determined representative genomes, each one representing one species (Mende *et al*, 2013), and using conservative thresholds for minimal sequencing depth and prevalence, we were able to confidently assess the variation in 71 abundant microbial species (Fig 1, Materials and Methods, and Table EV2). While the latter represent only a small fraction of known gut species, they account for the vast majority of the mapped reads (Table EV3; on average, 95.5% (SD = 6%) of assigned relative abundance per sample; Fig 1B).

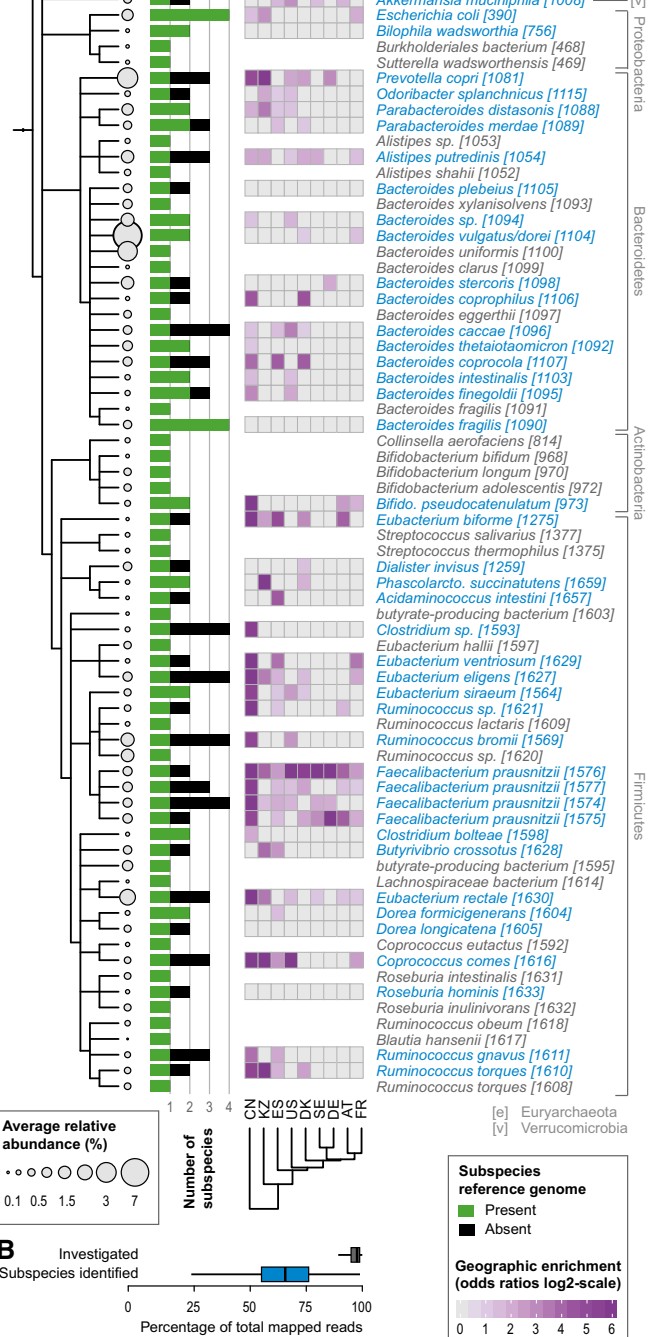

**Figure 1. Identification and prevalence of human gut microbial subspecies.**

A, B  Human gut microbial species explored for the existence of subspecies show wide phylogenetic spread according to NCBI taxonomy (A) and include *Methanobrevibacter smithii*, the main archaeal member of the human gut microbiome, as well as representatives of all abundant phyla. Species names are according to NCBI taxonomy, with species cluster (specI) identifiers according to Mende *et al* (2013), which splits some named species into multiple specI clusters. Of 71 investigated species, 44 stratify into subspecies (highlighted in blue). Each species' average abundance across 2,144 human gut metagenomes is proportional to the size of the circles on the cladogram. Bars represent the number of subspecies identified in each, with "1" indicating no subdivision. The black portion of the bar corresponds to subspecies for which no representative genome sequence is available from NCBI. Geographic enrichments of subspecies are displayed as a heat map (showing only significant enrichment, FDR-corrected Fisher test *P*-value < 0.05, per country as maximum log-odds ratio across conspecific subspecies). Subspecies with a restricted geographic range are predominantly found in the Chinese and Kazakh populations. The 71 investigated species captured an average of 95.5% of sequencing reads that were assigned to any reference genome. The subset of 44 species with identified subspecies accounted for the majority of this abundance (B).

## Subspecies delineation

To delineate population structure corresponding to subspecies (referred to as metagenomic subspecies, MGSS, in the following), we used metaSNV (Costea *et al*, 2017) to compute a matrix of genomic allele distances between all samples that allowed variant calling for any given species (Materials and Methods), and identified clusters by applying a very stringent cutoff for separation (Materials and Methods). With this operational definition, we found between two and four subspecies in 44 of the 71 species (accounting for 72% (SD = 15%) of the assigned relative abundance per sample), totaling 112 subspecies (Table EV2). The horizontal coverage over the analyzed genomes was generally above 70% and did not correlate with the existences of subspecies (Appendix Fig S1). We thus uncovered extensive population structure reflecting naturally occurring subspecies in the majority of abundant gut microbes.

The 71 gut species investigated represent 28 genera, for 25 of which population structure has never been assessed before (Materials and Methods). Of the other three, for which population genetic studies have been published, one, *Escherichia*, was previously described to have well-defined structure below the species level (Chakraborty *et al*, 2015). In comparison with the four subspecies of *E. coli* determined here, there is an exact correspondence, such that MGSS1, MGSS2, MGSS3, and MGSS4 recapitulate phylogroups B1, A, D, and B2, respectively (Materials and Methods, Table EV4). For the remaining two genera, a cohesive subspecies scheme has not been established, which precluded a comparison.

To investigate whether the stratification into subspecies is a potentially more broad feature of human-associated prokaryotes, we further assessed the genomic variation of 676 oral samples, collected from different sites within the oral cavity (buccal mucosa, tongue dorsum, and supragingival plaque) of multiple individuals. Subjecting them to the same procedure as the gut metagenomes, we were able to investigate the existence of subspecies in 62 additional named bacterial species. Of these, 18 could be delineated into two subspecies, which may be found in different habitats of the oral cavity. For example, *Granulicatella adiacens [1358]* MGSS1 is only found in plaque, while MGSS2 only lives in the buccal mucosa (Table EV5).

We then focused on the human gut metagenome for which a wealth of public datasets exist that could be utilized by our approach. Thus, for each of the subspecies identified in human gut species, we determined a set of unique SNPs that unambiguously identify it; the median number of such "genotyping" positions is 2,133 (with 1,142 and 3,989 being the $25^{th}$ and $75^{th}$ quantile, respectively). Using these positions, we could assign subspecies in a substantially expanded set of samples with low sequence coverage (on average, a 60% increase in the number of samples profiled per species; see Materials and Methods and Table EV2). We found that 47 of the 112 identified subspecies lack a representative reference genome, sometimes even the most prevalent ones (Fig 1A, Table EV2), as similarly noted by independent studies (Lloyd-Price *et al*, 2017; Truong *et al*, 2017). Hence, the metagenomic SNP profiles not only increased our power to investigate the relation between subspecies and host properties, but also allowed for subspecies identification and quantification in new samples.

## Subspecies biogeography

Having analyzed metagenomes from nine countries and three continents (China, Kazakhstan, Sweden, Denmark, Germany, Austria, France, Spain, and the United States), we could assess the global geographic distribution of each subspecies. While many subspecies appeared to be distributed without any recognizable geographic pattern, some did show striking regional enrichments (Figs 1A and EV1), consistent with a recently published, independent study (Truong *et al*, 2017). Overall, the most restricted geographic ranges across subspecies were observed in the Chinese samples, followed by the ones from Kazakhstan, while European and American samples appeared more similar in their gut subspecies compositions (Fig 1A). For example, for *Eubacterium rectale*, one subspecies (*E. rectale* MGSS3) was almost exclusive to Chinese samples (see also Truong *et al*, 2017), while the other two were found in all other countries, including neighboring Kazakhstan (Fig EV2). Strong geographic segregation might explain why associations between *E. rectale* and host physiology have often proven to be unstable when testing in different cohorts, as these subspecies could not be profiled previously (David *et al*, 2014). When comparing geographic ranges across bacterial taxa, members of the Firmicutes phylum showed significantly more geographic restrictions compared to all other phyla (Wilcoxon test *P*-value = 0.002). The regional enrichments among several *Firmicutes* species discovered here might reflect an adaptation to specific environmental factors, many of which are more prevalent in some geographic regions, but not exclusively found there. Alternatively, the observed structure could be the result of drift and differences in dispersal among gut microbial subspecies, though further work is needed to disentangle the two.

## Subspecies dominance and persistence in individuals

We next investigated whether subspecies occurrences are also restricted in individuals and how stable an individual's gut subspecies composition is over time. It has been previously shown that overall strain-level composition is generally stable (Schloissnig *et al*, 2013; Lloyd-Price *et al*, 2017; Truong *et al*, 2017), while upon perturbations, such as fecal microbiota transplantation, lasting changes in strain populations can occur (Li *et al*, 2016).

In order to study the population structure of conspecific subspecies, we recorded the frequency of each subspecies in each sample (Fig 2A–E). For the 44 species with substructure, we generally observed a clear dominance (one subspecies represents more than 90% of the combined abundance in any given individual), and in 83% of the samples even exclusivity, of one conspecific subspecies, with no apparent effect of sequencing depth on deducing subspecies exclusivity (see Materials and Methods and Appendix Fig S2) consistent with a recent independent report (Truong *et al*, 2017). This observation is in line with the frequency profiles observed in most samples, where the vast majority of alleles are "fixed", indicating the presence of only one dominating strain. Specifically, for 41 of the 44 species with substructure, over 75% of samples have 90% of assessed variants with frequencies below 5 or above 95% (Appendix Fig S3). Moreover, subspecies exclusion is also in line with ecological theory predicting that for closely related taxa, one outcompetes the others in the same ecological niche (Hardin, 1960). To analyze temporal stability of conspecific subspecies, we

considered individuals for which samples from multiple time points were available and tracked the number of times we observed a change in the dominant one. Considering 74 individuals from the United States, each sampled twice about 200 days apart (Huttenhower *et al*, 2012), as high as 94% of the dominant subspecies were the same at both time points, almost all of them being exclusive. Of the remaining 6%, the vast majority involved individuals that had more than one co-occurring subspecies at the earlier time point already (in these, subspecies replacement was 17 times more likely than in cases with a single subspecies at baseline, Fisher test $P = 2.2 \times 10^{-16}$), indicating that mixed subspecies populations are more fragile over time. This is supported by a more in-depth comparison of three individuals for which we have collected more than 20 samples over more than 2 years, and for which we could monitor 16–18 species across multiple time points spanning at least 1 year. For two individuals, we only found one incidence of switching in each. In both cases, the switching was observed where subspecies coexisted and the most abundant one changed. In the third individual, sampled over the same time period, seven out of the 18 species which we were able to track after an antibiotic intervention showed a subspecies replacement, supporting a strong and long-lasting impact of treatment on microbial composition (Voigt *et al*, 2015) accompanied by a switch of conspecific subspecies dominance (Fig 2).

Our observation of general dominance or exclusivity of a single subspecies adds another layer to gut microbial individuality. That it persists over time under normal conditions highlights the potential of targeted microbial interventions for achieving long-lasting effects once a desired strain is successfully "implanted" by manipulations such as fecal microbial transplantation (Li *et al*, 2016) or probiotic treatment.

## Subspecies functional characterization

In the absence of phenotypic information (Nichols *et al*, 2011), a way of inferring functional differences between distinct taxa is finding genes that are consistent within, but different between them. Applied to subspecies, we first identified for a given bacterial species core genes (species core, SC), consistently present in all strains, which should encode necessary functions performed by all strains (Vernikos *et al*, 2015). We further determined from the other (accessory) genes of that species a subspecies-specific core (SSSC). These SSSC genes are shared by all strains of that subspecies, but are distinct from those in any other conspecific subspecies. This set of genes thus encodes the consistent functional differences between them (Materials and Methods). Both SC and SSSC sets were inferred *de novo*, through a newly developed method based on the co-abundance concept (Materials and Methods), previously used to pool genes into metagenomic species independent of reference genomes (Nielsen *et al*, 2014). Using the SNP frequencies at the genotyping positions of each subspecies along with the overall species abundance, we could identify genes whose abundance consistently correlated with these (i.e., both SC and SSSC) with high accuracy (Appendix Fig S4).

For the functional characterization of subspecies, we annotated the genes to functional categories (Materials and Methods) and ascertained whether these were specific to either SC or SSSC, by testing for enrichment of functional annotations such as SEED (Overbeek *et al*, 2005) pathways, eggNOG (Huerta-Cepas *et al*,

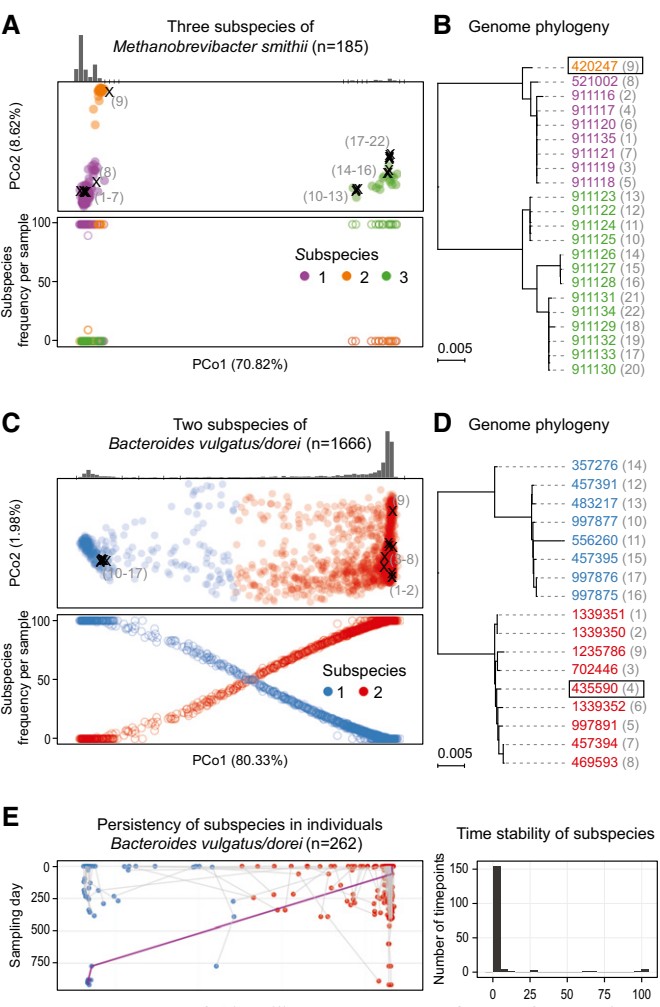

**Figure 2. Subspecies co-occurrence and phylogenetic consistency.**

A–D   Subspecies identified for *Methanobrevibacter smithii* and *Bacteroides vulgatus/dorei* are shown in principal coordinate (PCoA) projections of the between-sample distances based on single-nucleotide variations (see Materials and Methods). The first principal coordinate (PC) explains over 70% of the variation in both cases (panels A and C). Reference genomes have been projected into the same PCoA plots (marked with "×" in A and C; see Materials and Methods). The numbers adjacent to the placed genomes correspond to those shown in parentheses next to NCBI taxonomy identifiers (leaves) on the phylogenetic trees in (B, D), respectively. The sample density for each subspecies is highlighted by the histogram above, and the total number of samples in which the species could be quantified is indicated in headers. Quantification of the frequency of each subspecies (bottom plot in A and C) reveals that for *M. smithii*, only one sample has two subspecies co-occurring in one individual, while all the others have a single dominating one. In contrast, for *B. vulgatus/dorei*, co-occurrence is more commonly observed. Phylogenies reconstructed from the reference genomes (NCBI taxonomy identifiers; see Materials and Methods) are fully consistent with the SNP-based clustering. The representative genome for each species, relative to which genomic variants were called, is highlighted with a box.

E   In *B. vulgatus/dorei*, subspecies composition within each individual is generally stable over time, with a change of the dominant subspecies being rare even over a period of up to 1,000 days. An exception (highlighted by purple line) is seen for an individual, in which one dominant subspecies is replaced by another one after antibiotic treatment. The right-hand panel summarizes subspecies frequency changes, underlining remarkable stability over time.

2016b) categories, and KEGG (Kanehisa & Goto, 2000) modules in all the species with subspecies structure (Fig EV3). Compared to the combined SCs, the combined SSSCs were enriched in genes functionally related to phages (including genes for integration and excision), cell surface protein modifications (such as genes from the sortase pathway), chemotaxis, and motility as well as genes of unknown function (Table EV6). Such enrichments are broadly consistent with functional specialization of subspecies to different environmental niches. We next tested for functional differences between the SSSCs of different subspecies. Using homology-based KEGG annotation as well as SEED functional annotations across all species, we found 9 and 16, respectively, significant enrichments between conspecific subspecies, encoding a wide range of differences (at a cutoff of $P < 0.05$, corrected for multiple testing within each species; Table EV7). For example, we found significant differences in adhesion genes in two of four *E. coli* subspecies (MGSS3 and MGSS4), suggesting higher virulence potential (Lee *et al*, 2010). Using reference genomes and pathogenicity annotations from the PATRIC database (Wattam *et al*, 2014), we noted that these two subspecies were significantly enriched in isolates that caused diseases in the human host. These findings are in line with results on the corresponding phylogroups D and B2 (MGSS3 and MGSS4, respectively), which have been associated with extraintestinal infections in humans (Chakraborty *et al*, 2015). Moreover, metagenomic data from a recent enterohemorrhagic *E. coli* outbreak (Loman *et al*, 2013) indicates that the pathogenic strain comes from one of these two subspecies (Fig EV4). Even though our analysis is limited by the fact that an average of 70% of the SSSCs were not assigned to any functional category by either KEGG or SEED mappings, the example illustrates the utility of molecular gene content analysis to reveal differences in phenotypes.

**Subspecies associations with host phenotypes**

The differences in gene content and the long-term persistence of subspecies dominance in any individual prompted us to test whether conspecific subspecies differ in their correlations with host phenotypes. Indeed, we found five species with significant subspecies associations with microbial community diversity and two with host body mass index (BMI) (FDR-corrected *P*-value < 0.05, accounting for variation between studies; Table EV8 and Materials and Methods). In addition to these global associations, separate statistical tests within each study cohort revealed a strong negative association between type 2 diabetes status specifically in Chinese individuals (Qin *et al*, 2012) and *Bacteroides coprocola MGSS3*, which is enriched in healthy individuals (Fisher test *P*-value 2.98e-05). This association was, however, not significant in the two type 2 diabetes case–control studies with smaller cohorts (Karlsson *et al*, 2013; Le Chatelier *et al*, 2013).

The strongest subspecies association with host BMI was seen for *E. rectale*, which stratifies into three subspecies, one of which (MGSS3) is almost completely restricted to samples from China. All other humans studied were colonized by one of the other two subspecies, with 25% of samples containing a combination of the two. The observed stratification is in line with previous reports on the variation landscape of this species (Nayfach *et al*, 2016; Scholz *et al*, 2016), though no attempt at delineation or functional characterization was undertaken there. Our reconstruction of the SSSC,

as well as analysis of the coverage over the reference genome of this species (Fig 3A and C), revealed that *E. rectale* MGSS1 is missing at least 27 genes related to bacterial chemotaxis and flagellar assembly. These genes are necessary for bacterial motility and also represent an important signal for host immune activation, though both motility and the presence of these genes in the genome of different *Eubacterium* species are variable (Neville *et al*, 2013). The TLR5 receptor in human epithelial cells (colonocytes) recognizes the flagellum and induces a downstream cascade which results in initiation of pro-inflammatory pathways and secretion of IL-8 (Neville *et al*, 2013). Such low-grade inflammation has been repeatedly linked to obesity, increased insulin resistance, and diabetes (Gregor & Hotamisligil, 2011) providing a potential explanation for our observation that BMI and insulin resistance are significantly higher in individuals who predominantly harbor the flagellum-carrying subspecies (MGSS2 and MGSS3) in the Danish (Le Chatelier *et al*, 2013) and Swedish (Karlsson *et al*, 2013) cohorts (Fig 3D). This association is independent of the specific abundance of the subspecies in the individuals. Furthermore, individuals harboring subspecies MGSS2 and MGSS3 appeared to have lower community diversity (Fig 3D; Shannon diversity index), in line with previous observations of a negative correlation between community diversity and host BMI (Le Chatelier *et al*, 2013).

The four subspecies of the closely related *Eubacterium eligens* differ in similar ways (Fig 3B). At least nine genes related to bacterial chemotaxis and flagellar assembly are specific to *E. eligens* MGSS3. Only 7 and 10% of samples from the Danish (Le Chatelier *et al*, 2013) and the Swedish study contain this subspecies (14 and 4 samples, respectively, however not statistically significant due to small sample size). Such striking functional differences between subspecies and their associations with host phenotypes have implication for disease treatment, which could, for example, aim at replacing the "detrimental" subspecies by one without the respective trait. The examples illustrate that subspecies can reveal associations with host phenotypes that are undetectable at species level. In analogy, we hypothesize that at subspecies resolutions, the currently only weak and inconsistent co-occurrence patterns between gut microbial species across samples (Faust *et al*, 2012) should become much stronger, possibly revealing further links to host phenotypes and insights into the role of specific subspecies, rather than species, in maintaining gut microbial homeostasis.

# Discussion

We have shown here that the single-nucleotide variation landscape of most common gut commensals can be decomposed into cohesive units subordinate to species. These provide a meaningful grouping of microbial strains, which correspond to a more fine-grained level of variation that has been the focus of previous studies (Schloissnig *et al*, 2013; Nayfach *et al*, 2016; Lloyd-Price *et al*, 2017; Truong *et al*, 2017). In contrast to gut microbial strains, the vast majority (> 95%) of which is specific to an individual host (Truong *et al*, 2017), most subspecies can be detected ubiquitously in the global human population and only few appear to be geographically restricted.

Subspecies-specific gene content can be readily identified, enabling the exploration of the subspecies landscape of the human gut microbiome and its relation to the host. Given our stringent

operational definition of subspecies, we expect more to emerge with increasing sample numbers and sequencing depth. Their prevalence is likely to be generalizable to other habitats, and although only a few existing metagenomics datasets have sufficient depth for detailed analysis, we indeed find clear indications for it. For example, in the oral cavity, we observe many subspecies, often with

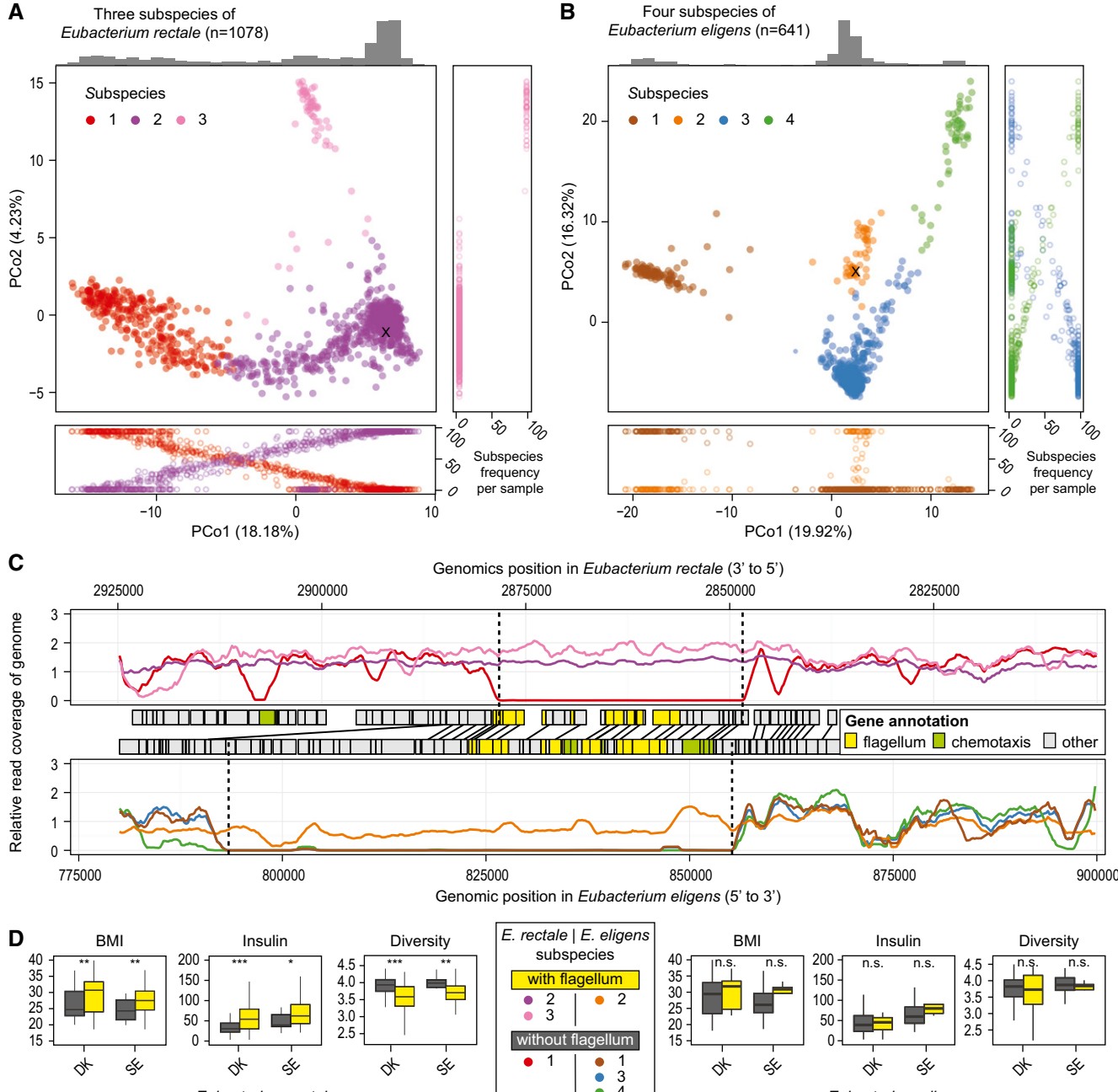

**Figure 3.** **Gene complement differences between subspecies and their potential implication for the host.**

A, B   Three subspecies of *Eubacterium rectale* (A) and four of *Eubacterium eligens* (B) are color-coded in PCoA plots. In most individuals, we observe dominance of one subspecies at a time (see frequency plots alongside PCoAs).

C    Functionally, the main distinction between some of the subspecies is large deletions which harbor many flagellum and chemotaxis-related genes (based on SEED as well as KEGG annotations), as evident from genomic read coverage (subspecies color-coded as in A and B).

D    Grouping *E. rectale* individuals from the Danish (DK) and Swedish (SE) studies, based on this deletion, shows a significant (*$P < 0.05$, **$P < 0.01$, ***$P < 0.001$) increase in BMI and blood fasting insulin levels as well as a decrease in overall community diversity in individuals who are predominantly colonized by the flagellum-carrying subspecies (see Materials and Methods). The same trend is observed for *E. eligens* subspecies in the same studies, though not statistically significant (n.s.). In the boxplots, the median is given as horizontal line and boxes represent the 25th and 75th percentiles. Whiskers extend to the point closest to 1.5 times the interquartile range.

apparent preferences for different sub-habitats, such as buccal mucosa or gingival plaque (Table EV5).

Although it is very difficult to prove phenotypic or ecological differences between the identified conspecific subspecies and we can only infer functional differences using gene pool reconstructions, our findings support the ecotype view of bacterial evolution, suggesting the existence of cohesive clusters of strains with the same ecological properties, each of them subject to their own evolutionary fate (Cohan, 2001). The partially distinct global dispersal patterns, functional repertoires, and associations with host properties imply that subspecies have an important role in ecology and represent a natural taxonomic level, which can be operationally defined using sequence data. We have shown the practical relevance of subspecies in the human gut microbiome, by their associations with host properties, which remain undetectable when only considering species (or lower resolution entities such as 16S OTUs or genera). As these associations are based on a limited set of genes, they also provide an entry point for mechanistic follow-up studies.

# Materials and Methods

## Data collection

We have used data from multiple published studies, as well as novel data collected for this work. Specifically, we used 239 stool samples from the HMP (Huttenhower *et al*, 2012) collected from 139 US American individuals, as well as 676 oral samples from the same individuals; 156 samples from as many Austrian individuals (Yu *et al*, 2015); 368 Chinese samples (Qin *et al*, 2012); 387 Danish samples from the MetaHIT cohort; 156 French and 40 German samples (Zeller *et al*, 2014); 126 novel German samples collected from 26 individuals; 172 novel Kazakh samples from 86 individuals; 359 Spanish samples from MetaHIT; and 145 Swedish samples (Karlsson *et al*, 2013; Table EV1). Informed consent was obtained from the German individuals through the my.microbes project (http://my.microbes.eu). The study protocol was approved by the respective institutional review board (EMBL Bioethics Internal Advisory Board) and is in agreement with the WMA Declaration of Helsinki and the Department of Health and Human Services Belmont Report. Informed consent was also obtained for all Kazakh participants before sample collection. The study protocol and consent documents were approved by the Ethics Committee of the Centre for Life Sciences National Laboratory Astana Nazarbayev University with ethical approval number 311/2537 (IORG0006963) as well as EMBL. All new samples were processed the same way, as described below.

## DNA isolation, library preparation, and metagenomic sequencing of novel samples

Genomic DNA was extracted from frozen fecal samples as previously described (Zeller *et al*, 2014) using the GNOME® DNA Isolation Kit (MP Biomedicals).

Library generation and shotgun sequencing were carried out on the Illumina HiSeq 2000/2500 (Illumina, San Diego, CA, USA) platform. All samples were paired-end-sequenced with 100 bp read lengths at the Genomics Core Facility, European Molecular Biology Laboratory. Novel samples have been submitted to NCBI under accession number PRJEB17632.

## Data preprocessing and species abundance estimation

All samples were processed with the same computational protocol. Reads were quality-filtered and screened against the human genome sequence for removing contamination as previously described (Zeller *et al*, 2014). Species abundance was calculated using established MOCAT (Kultima *et al*, 2012) protocols for specI clusters (Mende *et al*, 2013). Throughout the manuscript, we used specI clusters at the species level related via the NCBI taxonomy database as a taxonomic reference. Additionally, mOTU abundances were also determined using standard MOCAT procedures (Sunagawa *et al*, 2013), but exclusively used to estimate species diversity (Table EV1).

For calling genomic variants, all metagenomic sequencing reads were additionally mapped to a reference set consisting of 1,753 genomes (each representative of one specI cluster) (Mende *et al*, 2013), using MOCAT (Kultima *et al*, 2012) with default parameters. Specifically, reads were mapped at 97% identity and multiple mappers were discarded. Computation of genome coverage for each specI cluster was performed using qaCompute (https://github.com/CosteaPaul/qaTools), resulting in estimations of both horizontal and vertical coverage per sample, per genome.

## SNP calling

Population SNPs were called using metaSNV (Costea *et al*, 2017), which resulted in 19,221,237 positions over 1,753 genomes.

## Determining subspecies

Determination of subspecies structure proceeded through the following steps: Firstly, the set of samples considered for each species was restricted to a high-confidence discovery set (see below) to ensure accurate variant determination. Based on these variants, a distance was then computed between all samples and subspecies determined on this basis. Finally, variants specific to each subspecies (genotyping positions) were computed and used to expand subspecies assignments to new samples or ones that did not meet the criteria for inclusion in the discovery set.

## Discovery set

For avoiding issues caused by coverage variation, which hampers comparison between samples, we selected a specific subset of samples for the analysis of each species. For inclusion, we required vertical coverage higher or equal to 5× and horizontal coverage higher or equal to 40%. On this basis, we only considered species in our analysis for which at least 50 samples had sufficient coverage resulting in the set of 71 species reported. For each of these, variant positions were sub-selected such that only those with 5× coverage in at least half of the samples in the respective discovery set were kept, to ensure that only genomic variation over the core genome is compared across samples thus avoiding ascertainment bias due to missing values with non-random distribution. Variant positions

were eventually encoded as frequency of the non-reference allele in each of the samples.

## Clustering

To survey genomic (dis)similarity between conspecific strains carried by each individual, we used a modified Manhattan distance to assign a divergence of 0 to a pair of samples with an identical variation profile and 1 to completely different samples: $\sum_{i=1}^{n} |S1_i - S2_i|/n$, where $S1_i$ and $S2_i$ are the frequencies of SNV "$i$" in one and the other sample, respectively, and "$n$" is the total number of compared positions. Positions only covered in one of the samples were not considered in this distance computation, and at least 1,000 positions are required to compute a valid distance (i.e., $n > 1,000$). We used partitioning around medoids to determine clustering for any given number of clusters $k$, between 2 and 10. In this clustering step, we only kept one time-point sample for individuals which have been sampled more than once. Using the prediction strength (PS) (Tibshirani & Walther, 2005), we determined the support of the clustering for each $k$. The highest number of clusters that have a PS above 0.8 was considered to be the number of subspecies, as recommended by Tibshirani and Walther for determining high-quality clusters (Tibshirani & Walther, 2005). If there were no values above, we conservatively assumed this to be insufficient evidence for the existence of distinct subspecies.

## Reference genome placement

In order to place genomes into the variation space, we simulated reads from them (Mende *et al*, 2012) at 20× coverage and mapped these to the representative genome set at 97% identity, using the exact same parameters as when mapping metagenomic samples to the representative set. We then assessed variation over them in the same way we did for metagenomic samples, resulting in a variation profile that is completely comparable to the sample background. We were thus able to compute a distance from any sample to any genome that mapped to a given species.

## Genotyping positions

To determine a subset of variant positions, at which alleles allow for high-confidence distinction of subspecies, we selected positions where the mean allele frequency difference between one subspecies and the rest was greater than 80%. These are thus positions which occur at high frequency in one subspecies and not any of the others; we denote these as genotyping positions. To quantify subspecies in a new sample, we compute the median allele frequency over all genotyping positions for each subspecies (in exceptional cases, less than 5% of the time, where the individual subspecies frequencies add up to below 80% or to more than 120%, we do not assign any subspecies for that species). Benchmarking these assignments on samples from the discovery set, we did not observe any errors.

## Estimating subspecies abundances

Subspecies abundance was determined by multiplying the abundance of a species (via standard specI profiling; see above) with the median relative frequency determined for each subspecies based on the genotyping positions.

## Exclusivity of subspecies

Considering species which have more than 50× coverage in at least 50 samples, we illustrate the relation between coverage and abundance of the dominating subspecies in a sample (where this abundance is 100%, we say that subspecies is exclusively present) (Appendix Fig S2) and note that this indicates no relationship between coverage and dominance, even at coverages as high as 1,000×.

We further used a Fisher test to investigate whether there is a significant difference between the number of samples showing exclusively one subspecies, given a coverage above or below 50×. None of the observed differences were significant ($P$-value $\geq 0.05$, without any correction for multiple testing).

## Contextualizing metagenomic subspecies delineation

To obtain an overview of species whose pan-genome or population structure has been studied previously, we performed a systematic search of all PubMed titles and abstracts, including the following terms: pangenome, pan-genome, population, and phylogenetic; and the respective genus name. Manual curation of the search results was performed to exclude publications which were not about population structure of a species, but of structure within the genus.

The subspecies delineations proposed here can be used as a framework for future microbiological research. For example, a pan-genomic survey of capsule diversity in *Bacteroides fragilis* (Patrick *et al*, 2010) compared three strains belonging to MGSS1. That this is only one out of four *B. fragilis [1090]* subspecies detected here highlights the potential of our culture-independent metagenomics framework to inform and contextualize microbial (pan-)genomics studies.

## *Escherichia coli* phylogroup determination

We determined the phylogroups designation of each *E. coli* genome belonging to the specI cluster represented in the samples, by querying the Enterobase database (http://enterobase.warwick.ac.uk). Not all genomes had such an annotation and were thus not further considered. The annotation together with the subspecies designation for each genome is presented in Table EV4.

## Phylogenetic tree reconstruction

Phylogenetic trees (Fig 2A and B) were based on concatenated sequence alignments of 40 previously described universal marker genes (Creevey *et al*, 2011). They were constructed using the "sptree_raxml_all" workflow as implemented in ETE3 v3.0.0b36 (Huerta-Cepas *et al*, 2016a). In brief, multiple sequence alignments were built for the 40 marker proteins using Clustal Omega v1.2.1 (Sievers *et al*, 2011), translated into codon alignments, concatenated, and used to infer a phylogeny by RAxML v8.1.20 (Stamatakis, 2014).

## Subspecies gene content

To approximate the gene complement of each subspecies, we used a correlation-based inference which linked relative subspecies abundance (estimated as detailed above) to the relative abundance of genes directly identified from shotgun metagenomics datasets. To generate such gene profiles, we mapped all metagenomic reads to the integrated gene catalog (IGC) from Li *et al* (2014). Additionally, species abundance was correlated with the IGC. Genes with a Pearson correlation coefficient above 0.8 (computed on log10-transformed relative abundances) and a Spearman coefficient above 0.6 in either the subspecies correlations or the species ones were considered to belong to the taxa they correlated highest with. This resulted in a set of genes for the core of the species as well as a subspecies core genome, that is, genes that are unique to a given subspecies. To assess the accuracy of these assignments, we used five species for which reference genomes were available so that the gene complement of all subspecies was known and could be used for benchmarking. By mapping the determined species and subspecies cores to a collection of all genes from these benchmarking genomes, we could determine not only the best hit, but also all the ones above a given cutoff. This benchmarking indicates that the best-hit approach is more than 95% accurate (Appendix Fig S4). Annotation of the reconstructed gene complements was transferred from a published annotation of the IGC (Kultima *et al*, 2016). From this study, we specifically used annotations to eggNOG (Huerta-Cepas *et al*, 2016b), KEGG (Kanehisa & Goto, 2000), and SEED (Overbeek *et al*, 2005) as indicated in the main text.

## Host associations

To test for associations between subspecies abundance (within one species) and host phenotypes, such as type 2 diabetes status, we employed a testing framework in which we can account for batch effects between studies, by including "study" as a blocking factor [as implemented in the R coin package (Hothorn *et al*, 2006)]. Phenotyping data for all samples compared are available from Table EV1. For testing, we only included samples dominated by one specific subspecies (more than 95% abundant). *P*-values were FDR-corrected for multiple testing.

## Data and software availability

All samples used in this study are available from ENA, under the following accession numbers: PRJEB1220, PRJEB17632, PRJEB1786, PRJEB4336, PRJEB5224, PRJEB6070, PRJEB7774, PRJEB8347, PRJNA48479, SRP008047, and SRP011011. Specific sample identifiers are available in Table EV1.

Computer code for assessing nucleotide variation is available on the EMBL Git server, under https://git.embl.de/costea/metaSNV.

**Expanded View** for this article is available online.

## Acknowledgements

The authors are grateful to the EMBL Genomics Core Facility for sequencing services and Yan Ping Yuan and the EMBL Information Technology Core Facility for support with high-performance computing. They are moreover thankful to the members of the Bork group at EMBL for discussions and assistance. The work was supported by EMBL, CancerBiome (ERC-2010-AdG_20100317), Microbios (ERC-AdG-669830), MetaHIT (HEALTH-F4-2007-201052), and METACARDIS (FP7-HEALTH-2012-INNOVATION-I-305312). S.S. is supported by ETH Zurich and the Helmut Horten Foundation. FH received funding through the European Union's Horizon 2020 research and innovation program under the Marie Skłodowska-Curie grant agreement no. 660375. This work was supported by the BMBF-funded Heidelberg Center for Human Bioinformatics (HD-HuB) within the German Network for Bioinformatics Infrastructure (de.NBI, #031A537B).

## Author contributions

PIC performed the main analysis, interpreted the data, and wrote the manuscript. PB and GZ designed the study, interpreted the data, and wrote the manuscript. LPC, SS, RM, JH-C, KF, FH, and AK contributed to data analysis, data interpretation, and manuscript writing. All authors approved the final manuscript.

## Conflict of interest

The authors declare that they have no conflict of interest.

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
