## [Review Process File · Molecular Systems Biology]

Subspecies in the global human gut microbiome

Paul I. Costea, Luis Pedro Coelho, Shinichi Sunagawa, Robin Munch, Jamie Huerta-Cepas, Kristoffer Forslund, Falk Hildebrand, Almagul Kushugulova, Georg Zeller and Peer Bork

Review timeline:

Submission date:	29 April 2017
Editorial Decision:	4 July 2017
Revision received:	11 October 2017
Editorial Decision:	9 November 2017
Revision received:	24 November 2017
Accepted:	29 November 2017

Editor: Thomas Lemberger

Transaction Report:

1st Editorial Decision

4 July 2017

Thank you again for submitting your work to Molecular Systems Biology. I apologize again for the delay in getting back to you. We have now heard back from the two referees who agreed to evaluate your manuscript. As you will see from the reports below, the referees find the topic of your study of potential interest. They raise, however, substantial concerns on your work, which should be convincingly addressed in a revision of this work. Without repeating all the points raised by the reviewers, some of the major issues include the following:

- the aspects of novelty of the present study should be clarified
- a more rigorous and detailed account of the methodology used should be provided such that the analysis can be understood in detail. In particular the robustness of the approach should be demonstrated and thresholds or arbitrary cutoff should be made explicit and either justified or tested for their effect on the outcome of the analysis.

If you feel you can satisfactorily deal with these points and those listed by the referees, you may wish to submit a revised version of your manuscript. Please attach a covering letter giving details of the way in which you have handled each of the points raised by the referees. A revised manuscript will be once again subject to review and you probably understand that we can give you no guarantee at this stage that the eventual outcome will be favorable.

 REVIEWER REPORTS

Reviewer #1:

In this work, the authors use metagenomics and single nucleotide variations to investigate subspecies in the human microbiome. The authors collate a large collection of previously published metagenomes, in combination with some new metagenomes published as part of this study, and map

the reads to a large collection of reference genomes. They identify variant positions based on mapping to the reference genomes, and cluster these variant positions into "subspecies". They also link specific genes to specific subspecies based on co-abundance patterns. Finally, the authors correlate the abundance of specific subspecies with geographic areas, and in the case of *E. rectale*, with BMI as well.

There are a few concerns I have about this manuscript, the primary of which is its lack of novelty. Variant positions as seen by mapping human microbiome reads to reference genomes is something that had been studied many times, the most high profile of which was published by some of the same authors as this manuscript in 2012 (Schloissnig et al., 2012). This publication goes a step further and clusters these variant positions into subspecies, but this has been done many times previously (Donati et al., 2016; Erkus et al., 2013; Luo et al., 2015; Quince et al., 2016). This manuscript additionally correlates subspecies with accessory genes, but this too has been done many times previously (Quince et al., 2016; Scholz et al., 2016).

While this work appears to present a novel method of clustering SNPs into subspecies, they don't provide the details of how they do it (this point is discussed in greater detail below), and provide no evidence that their method is better than previous attempts at solving the problem. The other findings of the manuscript are geographic enrichment of subspecies, persistence of subspecies within an individual, and correlation of a subspecies of *E. rectale* with BMI, none of which are particularly novel or exciting.

Other specific points:

It's unclear how many samples are actually being used. Adding up the numbers reported in the methods gives $239 + 676 + 156 + 368 + 387 + 156 + 40 + 126 + 172 + 359 + 145 = 2,824$, but the supplemental table has 2,144 samples and the abstract lists 2,144 samples. Additionally, the title of the paper and other places specifically state that these are gut metagenomes, but the methods states that 676 American oral samples were used.

In the abstract and elsewhere it states that genes distinguishing subspecies were identified "in a manner that is independent of reference genomes". This is not true; as stated in the methods, this is performed based on an integrated gene catalogue generated from reference genomes.

In the abstract and elsewhere, relative abundance is reported in terms of "total assigned abundance". As far as I can tell, this corresponds to:

Reads mapping to genome set of interest / reads mapping to complete reference genome set

This is not a standard measure of relative abundance (as far as I know), and I'm not sure why it's useful. A much more widely used and useful measure would be regular relative abundance:

Reads mapping to genome set of interest / total reads in sample

I would encourage the replacement of all measures of "assigned relative abundance" with true relative abundance, or at the very least include both. This goes for Figure 1B as well.

There is substantial discussion in this paper about "subspecies biogeography", but I worry that the "regional enrichments" could be more due to differences between studies than biological differences. For example, in the data collection section, what are the differences in how the metagenomes were sequenced between studies? Are differences in sequencing technologies (extraction buffers, library preparation, ect.) and cohort selection (age, gender, disease status) accounted for? One could easily imagine these factors creating the false impression of regional enrichments.

The same comment as above applies to the section on "Subspecies associations with host phenotypes". Were differences between studies accounted for?

There are a number of methods that need clarifying, either in the main text or in a supplemental methods section. These include:

How was the reference genome set ascertained? How did you choose the representative genome of each specI cluster?

The SNP calling procedure seems prone to errors. As stated in the methods, a position is considered a SNP if it's above 1% frequency and there are more than 4 reads at the position. This means that at any coverage below 100x, a single read with a different base will result in the position being called a variant. Given that Illumina reads are somewhat error prone and non-specific mapping is an issue, a more sophisticated SNP-calling procedure may be in order

The methodology for determination of subspecies (a procedure at the crux of the paper) is unclear in a number of ways:

- When computing distance between samples, it reads: we used a modified Manhattan distance to assign a divergence of 0 to a pair of samples with an identical variation profile and 1 to completely different samples. Please state what the actual formula used for this calculation
- When computing distance between samples, it reads: Positions only covered in one of the samples were not considered in this distance computation and that only 40% horizontal coverage to state a genome is in a sample. Thus, there could easily be cases where a different 40% of the genome is covered in each sample, and so only a few variant positions are being compared between samples. Is the formula for computing distance robust to this issue?
- When clustering samples, it reads: We used partitioning around medoids to determine clustering for any given number of clusters k , between 2 and 10 and This highest number of clusters that has a PS above 0.8 was considered to be the number of subspecies. Why choose the highest number of subspecies with a prediction strength above 0.8? Why not chose the number of subspecies with the highest prediction strength? Is there any reason that 0.8 was chosen as a cutoff?
There is a statement: 47 of the 112 identified subspecies lack a representative reference genome. The methods has a section: Reference genome placement, but it provides no specifics for the thresholds used to determine if a reference genome is available (in terms of identity and horizontal coverage).

Given the complexity of the subspecies determination procedure, I feel a validation step would be very helpful. For a number of species with known subspecies complexity (*Klebsiella pneumoniae*, *E. faecalis*, *Pseudomonas auriginosa*, etc.), generate synthetic reads based on isolate genomes from different clades, mix them together in different proportions, and run the computational pipeline on them. Questions that could be answered are: How accurate is the procedure (in terms of recall and precision)? How different do reference genomes need to be (in terms of specI distance, ANI, 16S distance, ect.) in order to be considered different "subspecies" based on this procedure? Does this distance value change depending on the species?

Reviewer #2:

Costea et al. present a metagenome-based subspecies analysis of the human gut microbiome highlighting that key functional differences can be hidden within species. This is an important and timely message as the majority of the field is still using 16S rRNA to analyse the human microbiome, which will wholly miss the type of metabolic insights illustrated in this study. Overall the methods seem robust and key ones such as mOTUs and MOCAT, which are critical for the presented results, have already been published by the Bork group. However, the conclusion of subspecies exclusivity in 83% of samples (line 121) surely is a function of sampling depth, i.e. true absence of other conspecific subspecies vs that they were simply below detection. Sampling depth and estimated subspecies detection thresholds are not discussed beyond mention of 5x read coverage for inclusion of species in the study. Given that there are $\sim 10^{11}$ microbial cells per gram of human feces, and the datasets used in the study are likely < 10 Gb in size, sizable populations could be below detection, e.g. 10^7 cells per gram would be missed at this sequencing depth with a 5x coverage inclusion threshold. One way to confirm the argument of exclusivity would be to design haplotype-specific PCR primers for a few species and perform qPCR to estimate relative abundances of conspecific subspecies. Another issue related to co-occurring conspecific subspecies is that they should be able to recombine according to classical species definition. Was there really no evidence of homologous recombination between subspecies?

Minor comments.

Line 39. Prochlorococcus spelled incorrectly.

Line 153. Reference to isolates in this sentence could be confused with cultured isolates

Line 196. This should read MGSS1 to be consistent with the rest of the paragraph.

Line 202. The implication from this sentence is that the ancestral state for *E. rectale* is flagella-based motility. Is this an established fact? Do phylogenetic trees of flagella genes of *E. rectale* and related species / genera support a recent loss in MGSS1 as opposed to a recent gain in MGSS2 and 3?

Line 250. Only a passing mention of the 676 oral metagenomes in the Discussion (Table S7), not described in the Results at all.

Line 270. How much sequence data was generated? See discussion of detection threshold above.

Figure 1. There are four *Faecalibacterium prausnitzii* species presented in this figure. Has this been established in the literature? Also are the subspecies combined in the geographic enrichment heatmap? If yes, wouldn't it be more useful (and consistent with the papers main take home message) to show the subspecies enrichment separately?

1st Revision - authors' response

11 October 2017

Reviewer #1:

In this work, the authors use metagenomics and single nucleotide variations to investigate subspecies in the human microbiome. The authors collate a large collection of previously published metagenomes, in combination with some new metagenomes published as part of this study, and map the reads to a large collection of reference genomes. They identify variant positions based on mapping to the reference genomes, and cluster these variant positions into "subspecies". They also link specific genes to specific subspecies based on co-abundance patterns. Finally, the authors correlate the abundance of specific subspecies with geographic areas, and in the case of *E. rectale*, with BMI as well.

There are a few concerns I have about this manuscript, the primary of which is its lack of novelty. Variant positions as seen by mapping human microbiome reads to reference genomes is something that had been studied many times, the most high profile of which was published by some of the same authors as this manuscript in 2012 (Schloissnig et al., 2012). This publication goes a step further and clusters these variant positions into subspecies, but this has been done many times previously (Donati et al., 2016; Erkus et al., 2013; Luo et al., 2015; Quince et al., 2016). This manuscript additionally correlates subspecies with accessory genes, but this too has been done many times previously (Quince et al., 2016; Scholz et al., 2016).

Reply:

As the reviewer correctly points out, the present manuscript builds on methodology previously published, specifically work from the group on bacterial species definition based on a set of universal marker genes as well as determining variant positions on said species. The analysis presented here is a considerable addition to that originally assessed variation and our observation of the extent of subspecies existence and their properties constitute substantial novelty.

The reviewer is right in that clustering of variants into subspecies has recently been done in a few other studies. However, none of the studies were aware of and pointed to by the reviewer has applied it exhaustively to the human gut microbiome and reported on population structure of its member species. The mentioned studies applied clustering and anecdotally showed examples where it worked (and some where it did not) without a methodology to properly discriminate between the two scenarios in a systematic way. To make this explicit to the reader we have now rephrased the last paragraph of the introduction to read:

“Current genomic approaches to determine population structure are hampered by their dependence on bacterial isolates and sequenced genomes. Due to limited throughput and biased by growth requirements, cultivation-based assessments are unviable for the study of complex microbial communities. Recently, the advent of culture-independent metagenomic approaches and newly developed analysis tools have made it possible to study the population structure and genomic variation of microbes in their natural environment (15, 16). Here, we devised a novel approach for broadly delineating subspecies in the majority of abundant gut microbes and determined their associated single nucleotide variants in order to quantify and characterize them in their natural habitat. We observed consistent functional differences between subspecies with respect to their gene pools suggesting that they likely differ in phenotypic or ecological properties. We discovered that clearly delineated subspecies are the rule rather than the exception in the gut microbiome. When analyzing their global geographic distribution, we found that within a host they generally persist stably over time and are mutually exclusive. Illustrating the utility of the subspecies concept, we associated particular subspecies to specific genes and host phenotypes, with implications for disease. “

Moreover, there are important differences in our principled methodology which make this work novel. We apologize that this did not come across properly and thank the reviewer for pointing out specific instances where the level of detail is not satisfactory. In the revised manuscript we have considerably expanded on the methods and put renewed emphasis on important points throughout the manuscript. At this occasion we would also point out some features of the cited literature that are different from our approach to clarify the novelty of this manuscript:

1. Use of reference genomes:

Work by Donati et al. and Erkus et al., fundamentally depend on the existence of many assembled reference genomes for each bacterial species. Erkus et al. assemble such genomes from metagenomic data directly while the former uses existing genomes, which restricts it to very well characterized genera (specifically *Neisseria* in this case). Moreover, Donati et al. determine a set of informative single nucleotide polymorphisms (SNPs) based on alignment of all reference genomes and choose these as markers for disentangling strains present in the oral samples of interest (we note that this paper is not concerned with bacteria that exist in the human gut). In contrast, our approach to structure is relatively unguided and requires the existence of only one genome per named species for us to be able to quantify variation and identify subspecies. This feature of our approach is important, as illustrated by the amount of subspecies which do not have a sequenced representative genome (Figure 1). Thus, a stratification relying on reference genomes would be unable to quantify structure in more than half of the species in which we observe it.

2. Level of resolution:

Many studies to date (Quince et al. and Lou et al. included) have endeavored to reconstruct strains from metagenomic data. Specifically, Quince et al. present a tool for de-novo extraction of strain and reconstruction of their gene content. They apply this to samples containing an outbreak strain of *E. coli* and show that they are able to recover its genome and gene complement with 95.7% accuracy. In order to look at population structure, the authors of this study would have to assemble hundreds of genomes over thousands of samples and then apply the kinds of clustering that we did, or use a similar method for determining clustering. Our approach does not aim at strain-level resolution of reconstruction which is currently only possible for some species that have a good reference genome coverage. We are specifically interested in a “natural” level of clustering that can be applied to many species and that generically describes an evolutionary unit within which a lot less variation is observed than between groups.

We are aware that the distinctions made here were not clearly stated in the previous version of the manuscript and we have now added such information where pertinent and have also cited the literature highlighted by the reviewer in order to give the reader a better grasp of existing work.

While this work appears to present a novel method of clustering SNPs into subspecies, they don't provide the details of how they do it (this point is discussed in greater detail below), and provide no evidence that their method is better than previous attempts at solving the problem.

Reply:

We thank the reviewer for raising the point of methodological detail and regret that the previous version did not go far enough in describing the clustering and the determination of subspecies. We have amended the methods part and hope that the level of detail in the revised manuscript is appropriate (see Clustering section in Methods).

The other findings of the manuscript are geographic enrichment of subspecies, persistence of subspecies within an individual, and correlation of a subspecies of *E. rectale* with BMI, none of which are particularly novel or exciting.

Reply:

We regret that the reviewer is not excited by the properties of the subspecies, such as time-stability and geography. The latter, specifically, we believe to be of particular interest, as still little is known about the transmission patterns of different species. We were able to show here that only a small number of species are made up of subspecies that specifically localize to one country while for most there is global mixing.

The correlation with BMI we also find particularly novel. Up to this point in time, the relationship of species abundance to BMI has been investigated and some signals have been consistently recovered. Here, however, we show that within the same species, the presence (above our high threshold of 5x genome coverage) of one or the other of the subspecies (independent of abundance) is correlated to a high difference in BMI.

Other specific points:

It's unclear how many samples are actually being used. Adding up the numbers reported in the methods gives $239 + 676 + 156 + 368 + 387 + 156 + 40 + 126 + 172 + 359 + 145 = 2,824$, but the supplemental table has 2,144 samples and the abstract lists 2,144 samples. Additionally, the title of the paper and other places specifically state that these are gut metagenomes, but the methods states that 676 American oral samples were used.

Reply:

We thank the reviewer for noticing this inconsistency and apologize for the confusion it created. We have now added details to the oral samples consideration. 2144 stool sample were used for all the analysis presented in the study. The additional 676 American oral samples were only used as a confirmatory analysis to highlight the fact that subspecies exist in the oral environment too and that there they show the partitioning properties that one might expect, namely different subspecies in aerobic and anaerobic environments in the oral cavity. Moreover, this observation in the oral cavity highlights just how different subspecies can be and hinting at their relevance. As also noted by reviewer #2, this result is now clearly highlighted in the results section.

In the abstract and elsewhere it states that genes distinguishing subspecies were identified "in a manner that is independent of reference genomes". This is not true; as stated in the methods, this is performed based on an integrated gene catalogue generated from reference genomes.

Reply:

We thank the reviewer for this comment. While reference genomes were also included in the gene catalogue, it is largely built from gene sequences that are directly assembled from metagenomics samples. While it is true that we need at least one reference genome per species, our method is able to detect the existence and reconstructs the gene content of subspecies for which no reference genome exists. In benchmarks, we confirmed the ability of our method to accurately recover gene complements for subspecies for which reference genomes exist (Suppl. Figure 3). Based on these we conclude that also for subspecies without reference genomes, their gene complement can be recovered.

In the abstract and elsewhere, relativize abundance is reported in terms of "total assigned abundance". As far as I can tell, this corresponds to:

Reads mapping to genome set of interest / reads mapping to complete reference genome set

This is not a standard measure of relative abundance (as far as I know), and I'm not sure why it's useful. A much more widely used and useful measure would be regular relative abundance:

Reads mapping to genome set of interest / total reads in sample

I would encourage the replacement of all measures of "assigned relative abundance" with true relative abundance, or at the very least include both. This goes for Figure 1B as well.

Reply:

We thank the reviewer for brining focus on this issue. The numbers we report (in terms of how much of the "abundances" can be split into subspecies) are only relevant in the context of the number of reads that may be assigned to the genomes we use. We now state this limitation clearly and highlight how many of the reads generally map to the reference set (Supplementary Table 1), so that the reader may form an idea of how much of the total abundance each of the numbers may represent.

Furthermore, we note that the relative abundances of tools like MetaPhlAn (one of the leading tools in the field) are a lot closer to "reads mapping to genome of interest/total reads mapped to reference genomes" than to the normalization to the total number of reads. This is because only a very small portion of reads in a sample will map to clade-specific markers used for profiling and for the rest it is not known if they do not map because they belong to a region of the genome not considered or because they do not belong to any species not represented by reference genomes (and marker genes for profiling).

There is substantial discussion in this paper about "subspecies biogeography", but I worry that the "regional enrichments" could be more due to differences between studies than biological differences. For example, in the data collection section, what are the differences in how the metagenomes were sequences between studies? Are differences in sequencing technologies (extraction buffers, library preparation, ect.) and cohort selection (age, gender, disease status) accounted for? One could easily imagine these factors creating the false impression of regional enrichments.

Reply:

We thank the reviewer for this comment and note that the enrichments we are talking about are of haplotype presence and not of abundance differences. For example, for *E. rectale*, the result that subspecies 3 is enriched in the Chinese population means that in this population we observe a haplotype that we do not observe in any other population (i.e. there are thousands of fixed variants in this population). Thus, while we agree with the reviewer that any analysis based on relative (species) abundance is likely to be affected by differences in experimental protocols and participant demographics, it is much less likely that differences in DNA extraction protocols would give rise to thousands of erroneous allele counts.

The same comment as above applies to the section on "Subspecies associations with host phenotypes". Were differences between studies accounted for?

Reply:

As per the answer above, we are here comparing the presence of a certain haplotype and not relative abundance. Moreover, we mainly test differences within each study individually and take others as independent confirmation. When testing across studies, we use blocked tests from the "coin" package with study as a blocking factor which properly accounts for any study confounders in the statistical tests (see Methods, Host associations).

There are a number of methods that need clarifying, either in the main text or in a supplemental methods section. These include:

How was the reference genome set ascertained? How did you choose the representative genome of each specI cluster?

Reply:

We used the reference genomes as published by Mende et al. We have clarified this in the beginning of the “Species Delineation” section which now reads: “By mapping the respective reads to a set of 1753 previously determined representative genomes, each one representing one species (5) and using conservative thresholds for minimal sequencing depth and prevalence, we were able to confidently assess the variation in 71 abundant microbial species”

The SNP calling procedure seems prone to errors. As stated in the methods, a position is considered a SNP if it's above 1% frequency and there are more than 4 reads at the position. This means that at any coverage below 100x, a single read with a different base will result in the position being called a variant. Given that Illumina reads are somewhat error prone and non-specific mapping is an issue, a more sophisticated SNP-calling procedure may be in order

Reply:

We note that the 4 reads refer to four variant containing reads. So, it would have to be a position covered at 400x and the variant seen 4 times for a call to be made. Moreover, even if such positions do get randomly called, they will not result in stratification between samples. While we agree with the reviewer that a more sophisticated calling procedure may be in order for being confident about low frequency variants, the population structure we are observing is based on thousands of high frequency polymorphisms which cannot be explained by sequencing errors.

The methodology for determination of subspecies (a procedure at the crux of the paper) is unclear in a number of ways:

- When computing distance between samples, it reads: we used a modified Manhattan distance to assign a divergence of 0 to a pair of samples with an identical variation profile and 1 to completely different samples. Please state what the actual formula used for this calculation

Reply:

Added to Methods, Clustering, which now reads: “To survey genomic (dis-)similarity between conspecific strains carried by each individual, we used a modified Manhattan distance to assign a divergence of 0 to a pair of samples with an identical variation profile and 1 to completely different samples: $(\sum_{i=1}^n \frac{|S1_i - S2_i|}{n})$, where $S1_i$ and $S2_i$ are the frequencies of SNV “i” in one and the other sample and “n” is the total number of compared positions. Positions only covered in one of the samples were not considered in this distance computation and at least 1000 positions are required to compute a valid distance (i.e. $n > 1000$). “

- When computing distance between samples, it reads: Positions only covered in one of the samples were not considered in this distance computation and that only 40% horizontal coverage to state a genome is in a sample. Thus, there could easily be cases where a different 40% of the genome is covered in each sample, and so only a few variant positions are being compared between samples. Is the formula for computing distance robust to this issue?

Reply:

We thank the reviewer for pointing out this issue, which could theoretically happen. However, in practice, we require at least 1000 positions in order to consider a comparison valid (now added to Methods, see above). Moreover, in the majority of cases, far more than 40% of the genome is covered. We now highlight this and have added a Supplementary Figure 1 for the purpose.

- When clustering samples, it reads: We used partitioning around medoids to determine clustering for any given number of clusters k, between 2 and 10 and This highest number of clusters that has a PS above 0.8 was considered to be the number of subspecies. Why choose the highest number of subspecies with a prediction strength above 0.8? Why not chose the number of subspecies with the highest prediction strength? Is there any reason that 0.8 was chosen as a cutoff?

Reply:

We choose the number of clusters as recommended by the authors of the PS method: “This and other experiments suggest that we choose the optimal number of clusters \hat{k} to be the largest k such that $ps(k)$ is above some threshold. Experiments reported later in the article show that a

threshold in the range .8–.9 works for well separated clusters. We think of k as the largest number of clusters that can be reliably predicted in the dataset.” [Cluster Validation by Prediction Strength, Robert Tibshirani and Guenther Walther].

We have modified the methods section to read: “The highest number of clusters that has a PS above 0.8 was considered to be the number of subspecies, as recommended by Tibshirani and Walther for determining high quality clusters (47)”

There is a statement: 47 of the 112 identified subspecies lack a representative reference genome. The methods has a section: Reference genome placement, but it provides no specifics for the thresholds used to determine if a reference genome is available (in terms of identity and horizontal coverage).

Reply:

We thank the reviewer for this comment and apologize for the lack of clarity. We have now rephrased that section to read: “In order to place genomes into the variation space, we simulated reads from them (47) at 20x coverage and mapped these to the representative genome set at 97% identity, using the exact same parameters as when mapping metagenomic samples to the representative set.”

Thus, if when mapping all reference genomes available, to a representative and comparing the variation profile to the subspecies, none map to a specific subspecies, we say that that subspecies lacks a reference genome.

Given the complexity of the subspecies determination procedure, I feel a validation step would be very helpful. For a number of species with known subspecies complexity (*Klebsiella pneumoniae*, *E. faecalis*, *Pseudomonas aeruginosa*, etc.), generate synthetic reads based on isolate genomes from different clades, mix them together in different proportions, and run the computational pipeline on them. Questions that could be answered are: How accurate is the procedure (in terms of recall and precision)? How different do reference genomes need to be (in terms of specI distance, ANI, 16S distance, ect.) in order to be considered different "subspecies" based on this procedure? Does this distance value change depending on the species?

Reply:

We thank the reviewer for the proposed validation framework, but note that in the stool metagenomics data we are analyzing we generally observe only one dominant strain per sample. Thus, sampling reference genomes and considering one per sample and then calling SNPs, would necessarily result in a variation estimate comparable to simply calling SNPs on the alignment against the representative genome. Thus, in this framework, the recovery of known subspecies structure would only depend on the stringency of the prediction strength cutoff for confident clustering. At the moment we are using the stringent value of 0.8, but do not think calibrating this value based on a small number of species would yield more robust globally applicable cutoffs. Moreover, we are able to determine population structure for one of the best studied bacterial species, namely *E. coli* and show that the recovered subspecies are perfectly congruent with the *E. coli* phylogroups. We believe this to be validation of the accuracy of our determination.

Reviewer #2:

Costea et al. present a metagenome-based subspecies analysis of the human gut microbiome highlighting that key functional differences can be hidden within species. This is an important and timely message as the majority of the field is still using 16S rRNA to analyse the human microbiome, which will wholly miss the type of metabolic insights illustrated in this study. Overall the methods seem robust and key ones such as mOTUs and MOCAT, which are critical for the presented results, have already been published by the Bork group. However, the conclusion of subspecies exclusivity in 83% of samples (line 121) surely is a function of sampling depth, i.e. true absence of other conspecific subspecies vs that they were simply below detection. Sampling depth and estimated subspecies detection thresholds are not discussed beyond mention of 5x read coverage

for inclusion of species in the study. Given that there are $\sim 10^{11}$ microbial cells per gram of human feces, and the datasets used in the study are likely < 10 Gb in size, sizable populations could be below detection, e.g. 10^7 cells per gram would be missed at this sequencing depth with a 5x coverage inclusion threshold. One way to confirm the argument of exclusivity would be to design haplotype-specific PCR primers for a few species and perform qPCR to estimate relative abundances of conspecific subspecies.

Reply:

We thank the reviewer for raising this issue. Indeed, exclusivity may be too strong a term, given coverage limitation. However, when analyzing the dependence of exclusivity to coverage, we see no evidence that they are anticorrelated.

He have now changed the section in question to read: “For the 44 species with substructure, we generally observed a clear dominance (one subspecies represents more than 90% of the combined abundance in any given individual) and in 83% of the samples even exclusivity, of one con-specific subspecies. While the average coverage over the considered genomes is less than 10x, we note that the observed pattern of exclusion is still visible in species that are also very highly covered, in some samples up to 1000x (see Methods and Supplementary Figure 4).”

Moreover, we have added another section to the Methods which now reads: “Considering species which have more than 50x coverage in at least 50 samples, we illustrate the relation between coverage and abundance of the dominating subspecies in a sample (where this abundance is 100%, we say that subspecies is exclusively present) (Supplemenatry Figure X) and note that this indicates no relationship between coverage and dominance, even at coverages as high as 1000x.

We further use a Fisher test to investigate if there is a significant difference between the number of samples showing exclusively one subspecies, given a coverage above or below 50x. None of the observed differences were significant (p value ≥ 0.05 , without any correction for multiple testing).”

Another issue related to co-occurring conspecific subspecies is that they should be able to recombine according to classical species definition. Was there really no evidence of homologous recombination between subspecies?

Reply:

We thank the reviewer for raising this point and agree that it would be of interest to investigate the extent to which homologous recombination is observed between and within subspecies. However, given our approach for determining variant positions indicative of one subspecies or the other, we would select against regions likely to recombine (as the positions would be observed in multiple subspecies and would thus not distinguish them). This means we cannot use our determined positions to investigate the likelihood of recombination.

Moreover, considering the approach we have taken to determine genes that are specific to a subspecies, we do not have enough information here either to answer the question of recombination, for a similar reason to that above; genes that are likely to recombine would be classified as “common core” by our methodology. Because of these properties of our analytical approach, we do not think our data allows us to answer the recombination question.

Minor comments.

Line 39. Prochlorococcus spelled incorrectly.

Reply:

We thank the reviewer for spotting this mistake and have now fixed it.

Line 153. Reference to isolates in this sentence could be confused with cultured isolates

Reply:

We agree with the reviewer that this is prone to causing confusion, so have replaced “isolates” with “strains”

Line 196. This should read MGSS1 to be consistent with the rest of the paragraph.

Reply:

We thank the reviewer for this comment, but note that MGSS3 is correct. This is the subspecies that is almost exclusively present in Chinese individuals, but it also contains the flagellum operon (Fig.3).

Line 202. The implication from this sentence is that the ancestral state for *E. rectale* is flagella-based motility. Is this an established fact? Do phylogenetic trees of flagella genes of *E. rectale* and related species / genera support a recent loss in MGSS1 as opposed to a recent gain in MGSS2 and 3?

Reply:

We thank the reviewer for this comment. We did not perform a comprehensive analysis of the Eubacteria phylogeny. However, based on findings from Neville et al. (<https://doi.org/10.1371/journal.pone.0068919>), who show that a similar flagellum operon is found in multiple Eubacterium species as well as in closely related Roseburia, we infer that this is likely a loss event. We do note that there is considerable observed variation in motility within the Eubacterium genus and have updated the text to reflect this.

Line 250. Only a passing mention of the 676 oral metagenomes in the Discussion (Table S7), not described in the Results at all.

Reply:

We agree with the reviewer that this finding deserves more focus and have thus expanded on it in the results section.

Line 270. How much sequence data was generated? See discussion of detection threshold above.

Reply:

A comprehensive table describing the sequencing depth per sample has been added as a supplementary table. Moreover, we now also report a per genome per sample average coverage table, for the 71 species considered in the analysis (Suppl. Table 3).

Figure 1. There are four *Faecalibacterium prausnitzii* species presented in this figure. Has this been established in the literature?

Reply:

We apologize that this was not made clear enough in the text. We are here using specI definitions of species, which in some cases split known species into multiple ones. The original publication by Mende et al. (Nat. Methods 2013) highlights several cases in which this makes sense taxonomically and in general, this consistent, operational definition appears to reflect taxonomy very well. Thus, a few species including *F. prausnitzii* were represented by multiple representative genomes. A note of clarification about this was added to the figure caption.

Also are the subspecies combined in the geographic enrichment heatmap? If yes, wouldn't it be more useful (and consistent with the papers main take home message) to show the subspecies enrichment separately?

Reply:

We thank the reviewer for this comment. Indeed, all geographic signal is aggregated in this table. We agree that a per subspecies geography is also of interest, so we have added a supplementary figure (Suppl. Figure 2) illustrating the geographical enrichment in such a way that it can be determined which subspecies is enriched in which country.

Thank you again for submitting your revised work to Molecular Systems Biology. We are now satisfied with the modifications made and we will be able to accept your manuscript pending the following minor points:

- please include a short discussion of your results in the light of the recent paper by Lloyd-Price et al (Nature 550:61 doi:10.1038/nature23889) and Truong et al (Genome Res 27:626).
- please include a running title, key words (up to 5), author contributions to the main manuscript
- for the HTML version of the paper, we would need:
 1. three to four 'bullet points' highlighting the main findings of your study
 2. a short 'blurb' text summarizing in two sentences the study (max. 250 characters)
 3. a 'thumbnail image' (width=211 x height=157 pixels, Illustrator, PowerPoint, OmniGraffle or jpeg format), which can be used as 'visual title' for the synopsis section of your paper.
- please include an author contributions statement after the Acknowledgements section (see <http://msb.embopress.org/authorguide>)
- please complete the CHECKLIST available at http://embopress.org/sites/default/files/Resources/EP_Author_Checklist_Master.xlsx. Please note that the Author Checklist will be published alongside the paper as part of the transparent process <http://msb.embopress.org/authorguide#transparentprocess>.
- please note that corresponding authors are required to supply an ORCID ID for their name upon submission of a revised manuscript (EMBO Press signed a joint statement to encourage ORCID adoption) <http://msb.embopress.org/authorguide#editorialprocess>.
- the 'Result' section should be labelled as such.
- Rename Methods section to Materials and Methods
- Figure legends need to be moved from the figures to the end of the main word file
- We need individual high resolution figure files.
- The reference list and callouts should be formatted in the MSB format.
- The legends from the supp table should be included in individual tabs in the excel sheets, so that people downloading the files can understand what the file is about.
- Figure 2C-E and 3A & C have to be explicitly called out from the text.
- The new data collected in the context of this work should be deposited in an appropriate repository.
- The software and the new data collected in this work should be made available and listed under a Data and Software Availability section, placed after Materials & Methods and following the example:

"Data and Software availability section:

- The datasets and computer code produced in this study are available in the following databases:
 - RNA-Seq data: Gene Expression Omnibus GSE46843
[<https://www.ncbi.nlm.nih.gov/geo/query/acc.cgi?acc=GSE46843>]
 - Chip-Seq data: Gene Expression Omnibus GSE46748
[<https://www.ncbi.nlm.nih.gov/geo/query/acc.cgi?acc=GSE46748>]
 - Protein interaction AP-MS data: PRIDE PXD000208
[<http://www.ebi.ac.uk/pride/archive/projects/PXD000208>]
 - Imaging dataset: Image Data Resource doi:10.17867/10000101
[<http://doi.org/10.17867/10000101>]
 - Modeling computer scripts: GitHub
[<https://github.com/SysBioChalmers/GECKO/releases/tag/v1.0>]
 - Metagenomic samples: European Nucleotide Archive list of accession numbers provided in Dataset EVxx
[data type]: [full name of the resource] [accession number/identifier] ([doi or URL or identifiers.org/DATABASE:ACCESSION])

The authors made the suggested editorial changes and submitted the final version of their manuscript.

Corresponding Author Name: Peer Bork

Manuscript Number: MSB-17-7589R